# CAX: CELLULAR AUTOMATA ACCELERATED IN JAX

**Maxence Faldor**
Department of Computing
Imperial College London
London, United Kingdom
m.faldor22@imperial.ac.uk

**Antoine Cully**
Department of Computing
Imperial College London
London, United Kingdom
a.cully@imperial.ac.uk

## ABSTRACT

Cellular automata have become a cornerstone for investigating emergence and self-organization across diverse scientific disciplines. However, the absence of a hardware-accelerated cellular automata library limits the exploration of new research directions, hinders collaboration, and impedes reproducibility. In this work, we introduce CAX (Cellular Automata Accelerated in JAX), a high-performance and flexible open-source library designed to accelerate cellular automata research. CAX delivers cutting-edge performance through hardware acceleration while maintaining flexibility through its modular architecture, intuitive API, and support for both discrete and continuous cellular automata in arbitrary dimensions. We demonstrate CAX's performance and flexibility through a wide range of benchmarks and applications. From classic models like elementary cellular automata and Conway's Game of Life to advanced applications such as growing neural cellular automata and self-classifying MNIST digits, CAX speeds up simulations up to 2,000 times faster. Furthermore, we demonstrate CAX's potential to accelerate research by presenting a collection of three novel cellular automata experiments, each implemented in just a few lines of code thanks to the library's modular architecture. Notably, we show that a simple one-dimensional cellular automaton can outperform GPT-4 on the 1D-ARC challenge.

## 1  INTRODUCTION

Emergence is a fundamental concept that has captivated thinkers across various fields of human inquiry, including philosophy, science and art (Holland, 2000). This fascinating phenomenon occurs when a complex entity exhibits properties that its constituent parts do not possess individually. From the collective intelligence of ant colonies to the formation of snowflakes, self-organization and emergence manifest in myriad ways. The study of self-organization and emergence holds the promise to unravel deep mysteries, from the origin of life to the development of conciousness.

Cellular Automata (CA) are models of computation that exemplify how complex patterns and sophisticated behaviors can arise from simple components interacting through basic rules. Originating from the work of Ulam and von Neumann in the 1940s (Neumann & Burks, 1966), these systems gained prominence with Conway's Game of Life in the 1970s (Gardner, 1970) and Wolfram's systematic studies in the 1980s (Wolfram, 2002). The discovery that even elementary cellular automata can be Turing-complete underscores their expressiveness (Cook, 2004). CAs serve as a powerful abstraction for investigating self-organization and emergence, offering insights into complex phenomena across scientific domains, from physics and biology to computer science and artificial life.

In recent years, the integration of machine learning techniques with cellular automata has opened new avenues for research in morphogenesis (Mordvintsev et al., 2020) and self-organization (Randazzo et al., 2020; 2021; Najarro et al., 2022). Neural Cellular Automata (NCA) represent a key advancement in this field, using differentiable neural networks to learn update rules through gradient-based optimization rather than relying on manual design. Recent works have combined NCA with advanced architectures like Convolutional Neural Networks (Pajouheshgar et al., 2023), Graph Neural Networks (Grattarola et al., 2021) and Vision Transformers (Tesfaldet et al., 2022). The convergence of machine learning and cellular automata research not only deepened our understanding of complex systems but also underscored the growing computational demands of CA experiments.

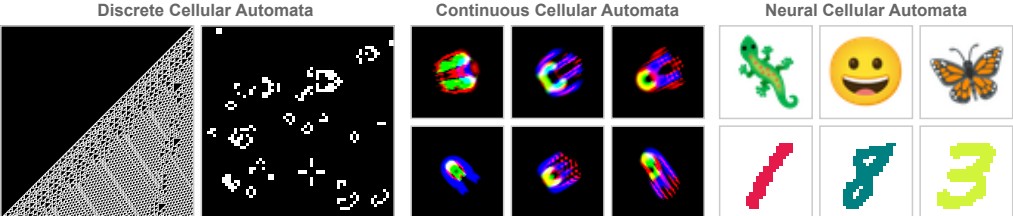

Figure 1: Cellular Automata types supported in CAX.

Despite their conceptual simplicity, cellular automata simulations can be computationally intensive, especially when scaling to higher dimensions with large numbers of cells or implementing backpropagation through time for NCAs. Moreover, the implementation of CA in research settings has often been fragmented, with individual researchers frequently reimplementing basic functionalities, creating custom implementations across various deep learning frameworks such as TensorFlow, JAX, and PyTorch. As the field continues to grow and attract increasing interest, there is a pressing need for a unified, robust library that facilitates collaboration, reproducibility, fast experimentation and exploration of new research directions.

In response to these challenges and opportunities, we present CAX: Cellular Automata Accelerated in JAX, an open-source library with cutting-edge performance, designed to provide a flexible and efficient framework for cellular automata research. CAX is built on JAX (Bradbury et al., 2018), a high-performance numerical computing library, enabling to speed up cellular automata simulations through massive parallelization across various hardware accelerators such as CPUs, GPUs, and TPUs. CAX is flexible and supports both discrete and continuous cellular automata with *any* number of dimensions, accommodating classic models like elementary cellular automata and Conway's Game of Life, as well as modern variants such as Lenia and Neural Cellular Automata (Figure 1).

JAX offers efficient vectorization of CA rules, enabling millions of cell updates to be processed simultaneously. It also provides automatic differentiation capabilities to backpropagate through time efficiently, facilitating the training of Neural Cellular Automata. CAX can run experiments with millions of cell updates in minutes, reducing computation times by up to 2,000 times compared to traditional implementations in our benchmark. This performance boost opens up new possibilities for large-scale CA experiments that were previously computationally prohibitive.

CAX's flexibility and potential to accelerate research is showcased through three novel cellular automata experiments. Thanks to CAX's modular architecture, each of these experiments is implemented in just a few lines of code, significantly reducing the barrier to entry for cellular automata research. Notably, we show that a simple one-dimensional cellular automaton implemented with CAX outperforms GPT-4 on the 1D-ARC challenge (Xu et al., 2024), see Section 5.3. Finally, to support users and facilitate adoption, CAX comes with high-quality, diverse examples and comprehensive documentation. The list of implemented CAs is detailed in Table 1.

## 2 BACKGROUND

### 2.1 CELLULAR AUTOMATA

A *cellular automaton* is a simple model of computation consisting of a regular grid of cells, each in a particular state. The grid can be in any finite number of dimensions. For each cell, a set of cells called its neighborhood is defined relative to the specified cell. The grid is updated at discrete time steps according to a fixed rule that determines the new state of each cell based on its current state and the states of the cells in its neighborhood.

A CA is defined by a tuple $(\mathcal{L}, \mathcal{S}, \mathcal{N}, \phi)$, where $\mathcal{L}$ is the $d$-dimensional *lattice* or *grid* with $c$ channels, $\mathcal{S}$ is the *cell state set*, $\mathcal{N} \subset \mathcal{L}$ is the *neighborhood* of the origin, and $\phi : \mathcal{S}^{\mathcal{N}} \to \mathcal{S}$ is the *local rule*. A mapping from the grid to the cell state set $\mathbf{S} : \mathcal{L} \to \mathcal{S}$ is called a *configuration* or *pattern*. In this work, we will simply refer to it by the *state* of the CA. $\mathbf{S}(\mathbf{x})$ represents the state of a cell $\mathbf{x} \in \mathcal{L}$. Additionally, we denote the neighborhood of a cell $\mathbf{x} \in \mathcal{L}$ by $\mathcal{N}_{\mathbf{x}} = \{\mathbf{x} + \mathbf{n}, \mathbf{n} \in \mathcal{N}\}$, and $\mathbf{S}(\mathcal{N}_{\mathbf{x}}) = \{\mathbf{S}(\mathbf{n}), \mathbf{n} \in \mathcal{N}_{\mathbf{x}}\}$.

The *global rule* $\Phi : \mathcal{S}^{\mathcal{L}} \to \mathcal{S}^{\mathcal{L}}$ applies the local rule uniformly to all cells in the lattice and is defined such that, for all $\mathbf{x}$ in $\mathcal{L}$, $\Phi(\mathbf{S})(\mathbf{x}) = \phi(\mathbf{S}(\mathcal{N}_{\mathbf{x}}))$. A cellular automaton is initialized with a state $\mathbf{S}_0$. Then, the state is updated according to the global rule $\Phi$ at each discrete time step $t \in \mathbb{N}$, to give,

$$\mathbf{S}_1 = \Phi(\mathbf{S}_0), \mathbf{S}_2 = \Phi(\mathbf{S}_1), \ldots$$

The close connection between CA and recurrent convolutional neural networks has been observed by numerous researchers (Gilpin, 2019; Wulff & Hertz, 1992; Mordvintsev et al., 2020; Chan, 2020). For example, the general NCA architecture introduced by Mordvintsev et al. (2020) can be conceptualized as a "recurrent residual convolutional neural network with per-cell dropout".

## 2.2 CONTROLLABLE CELLULAR AUTOMATA

A *controllable cellular automaton* (CCA) is a generalization of CA that incorporates the ability to accept external inputs at each time step. CCAs formalize the concept of Goal-Guided NCA that has been introduced in the literature by Sudhakaran et al. (2022). The external inputs can modify the behavior of CCAs, offering the possibility to respond dynamically to changing conditions or control signals while maintaining the fundamental principles of cellular automata.

A CCA is defined by a tuple $(\mathcal{L}, \mathcal{S}, \mathcal{I}, \mathcal{N}, \phi)$, where $\mathcal{I}$ is the *input set* and $\phi : \mathcal{S}^{\mathcal{N}} \times \mathcal{I}^{\mathcal{N}} \to \mathcal{S}$ is the *controllable local rule*. A mapping from the grid to the input set $\mathbf{I} : \mathcal{L} \to \mathcal{I}$ is called the *input*. $\mathbf{I}(\mathbf{x})$ represents the input of a cell $\mathbf{x} \in \mathcal{L}$. Similarly to the state, we denote $\mathbf{I}(\mathcal{N}_{\mathbf{x}}) = \{\mathbf{I}(\mathbf{n}), \mathbf{n} \in \mathcal{N}_{\mathbf{x}}\}$.

The *controllable global rule* $\Phi : \mathcal{S}^{\mathcal{L}} \times \mathcal{I}^{\mathcal{L}} \to \mathcal{S}^{\mathcal{L}}$ is defined such that, for all $\mathbf{x}$ in $\mathcal{L}$, $\Phi(\mathbf{S}, \mathbf{I})(\mathbf{x}) = \phi(\mathbf{S}(\mathcal{N}_{\mathbf{x}}), \mathbf{I}(\mathcal{N}_{\mathbf{x}}))$. A controllable cellular automaton is initialized with an initial state $\mathbf{S}_0$. Then, the state is updated according to the controllable global rule $\Phi$ and a sequence of input $(\mathbf{I}_t)_{t \geq 0}$ at each discrete time step $t \in \mathbb{N}$, to give,

$$\mathbf{S}_1 = \Phi(\mathbf{S}_0, \mathbf{I}_0), \mathbf{S}_2 = \Phi(\mathbf{S}_1, \mathbf{I}_1), \ldots$$

As discussed in Section 2.1, CAs can be conceptualized as recurrent convolutional neural networks. However, traditional CAs lack the ability to take external inputs at each time step. CCAs extend the capabilities of traditional CAs by making them responsive to external inputs, akin to recurrent neural networks processing sequential data. CCAs bridge the gap between recurrent convolutional neural networks and cellular automata, opening up new possibilities for modeling complex systems that exhibit both autonomous emergent behavior and responsiveness to external control.

## 2.3 RELATED WORK

The field of CA has spawned numerous tools and libraries to support research and experimentation, with CellPyLib (Antunes, 2021) emerging as one of the most popular and versatile options. This Python library offers a simple yet powerful interface for working with 1- and 2-dimensional CA, supporting both discrete and continuous states, making it an ideal baseline for comparative studies and further development. While it provides implementations of classic CA models like Conway's Game of Life and Wireworld, CellPyLib is not hardware-accelerated and does not support the training of neural cellular automata. Golly is a cross-platform application for exploring Conway's Game of Life and many other types of cellular automata. Golly's features include 3D CA rules, custom rule loading, and scripting via Lua or Python. While powerful and versatile for traditional CA, Golly is not designed for hardware acceleration or integration with modern machine learning frameworks.

The recent surge in artificial intelligence has increased the availability of computational resources, and encouraged the development of sophisticated tools such as JAX (Bradbury et al., 2018), a high-performance numerical computing library with automatic differentiation and JIT compilation. A rich ecosystem of specialized libraries has emerged around JAX, such as Flax (Heek et al., 2024) for neural networks, RLax (DeepMind et al., 2020) for reinforcement learning, and EvoSax (Lange, 2022), EvoJax (Tang et al., 2022) and QDax (Chalumeau et al., 2023) for evolutionary algorithms.

In the realm of cellular automata, there have been efforts to implement specific CA models using JAX. For instance, EvoJax (Tang et al., 2022) and Leniax (Giraud, 2022) both provide a hardware-accelerated Lenia implementation. Biomaker CA (Randazzo & Mordvintsev, 2023), a specific CA model focusing on biological pattern formation, further demonstrates the potential of JAX in CA

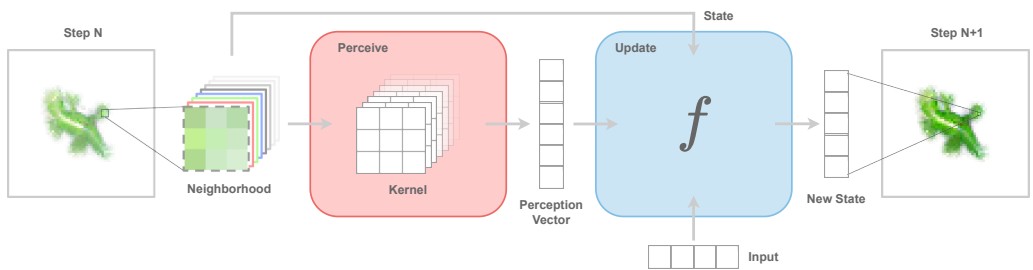

Figure 2: High-level architecture of CAX, illustrating the modular design with **perceive** and **update** modules. This flexible structure supports various CA types across multiple dimensions. (Adapted from Mordvintsev et al. (2020) under CC-BY 4.0 license.)

research. Finally, various GitHub repositories replicate results from neural cellular automata papers, but these implementations are typically narrow in focus. Recent advancements in continuous cellular automata research have also benefited from JAX-based implementations. These include Lenia (Chan, 2020) and Leniabreeder (Faldor & Cully, 2024), which have enabled large-scale simulations of open-ended evolution in continuous cellular automata (Chan, 2023).

While existing implementations demonstrate JAX's potential in CA research, they also reveal significant gaps in the field. Current tools are often specialized for specific CA types (e.g., discrete, 1- and 2-dimensional), narrow in focus (e.g., replicating specific neural CA papers), or lack hardware acceleration. This limitation underscores the need for a comprehensive, flexible, and efficient library that can handle a broad spectrum of CA types while leveraging hardware acceleration. CAX aims to address this gap by providing a versatile, JAX-based tool to accelerate progress across the entire landscape of cellular automata research.

## 3 CAX: CELLULAR AUTOMATA ACCELERATED IN JAX

CAX is a high-performance and flexible open-source library designed to accelerate cellular automata research. In this section, we detail CAX's architecture, design and key features. At its core, CAX leverages JAX and Flax (Heek et al., 2024), capitalizing on the well-established connection between CA and recurrent convolutional neural networks. This synergy, discussed in Section 2, allows CAX to harness advancements in machine learning to accelerate CA research. CAX offers a modular and intuitive design through a user-friendly interface, supporting both discrete and continuous cellular automata across any number of dimensions. This flexibility enables researchers to seamlessly transition between different CA within a single, unified framework (Table 1). We invite readers to experience CAX's capabilities firsthand by accessing our examples as interactive notebooks in Google Colab, conveniently linked in the README of the repository https://github.com/maxencefaldor/cax.

### 3.1 ARCHITECTURE AND DESIGN

CAX introduces a unifying framework for all cellular automata types. This flexible architecture is built upon two key components: the **perceive** module and the **update** module, see Figure 2. Together, these modules define the local rule of the CA. At each time step, this local rule is applied uniformly to all cells in the grid, generating the next global state of the system, as explained in Section 2.1. This modular approach not only provides a clear separation of concerns but also facilitates easy experimentation and extension of existing CA models.

Building on the controllable CA framework introduced in Section 2.2, our architecture generalizes the neural cellular automata approach to recurrent convolutional neural architectures, enabling seamless integration of external inputs and control signals. By implementing both modules using standard machine learning components from Flax, CAX makes it straightforward to experiment with various neural architectures while maintaining the cellular automata paradigm - from simple convolutional layers to sophisticated attention mechanisms.

```python
@nnx.jit
def step(self, state: State, input: Input | None = None) -> State:
    """Perform a single step of the CA.

    Args:
        state: Current state.
        input: Optional input.

    Returns:
        Updated state.

    """
    perception = self.perceive(state)
    state = self.update(state, perception, input)
    return state
```

The architecture of CAX allows for easy composition of different perceive and update modules, enabling the creation of a wide variety of cellular automata models. This modular design also facilitates experimentation with new types of cellular automata by allowing users to define custom perceive and update modules while leveraging the existing infrastructure provided by the library.

### 3.1.1 PERCEIVE MODULE

The perceive module in CAX is responsible for gathering information from the neighborhood of each cell. This information is then used by the update module to determine the cell's next state. CAX provides several perception mechanisms, including Convolutional Perception, Depthwise Convolutional Perception and Fast Fourier Transform Perception. The perceive modules are designed to be flexible and can be customized for different types of cellular automata.

### 3.1.2 UPDATE MODULE

The update module in CAX is responsible for determining the next state of each cell based on its current state and the information gathered by the perceive module. CAX provides several update mechanisms, including MLP Update, Residual Update and Neural Cellular Automata Update. Like the perceive modules, the update modules are designed to be flexible and can be customized for different cellular automata models.

## 3.2 FEATURES

### 3.2.1 PERFORMANCE

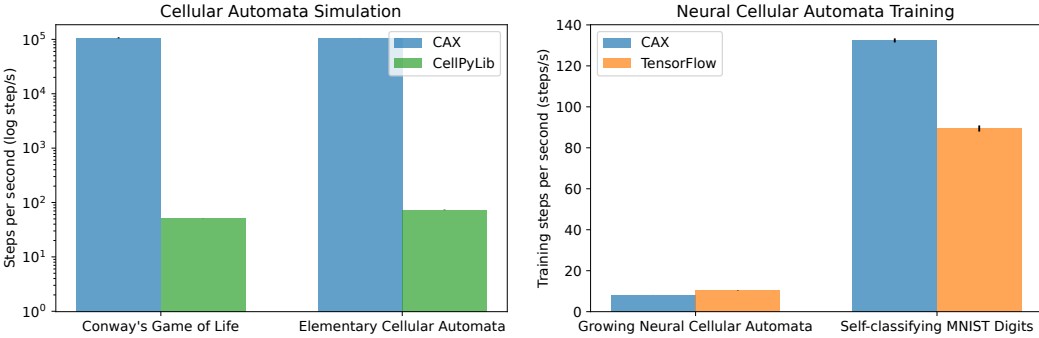

Figure 3: Performance benchmarks of CAX. **Left:** Simulation speed comparison between CAX and CellPyLib for classical cellular automata. CAX demonstrates a 1,400x speed-up for Elementary Cellular Automata and a 2,000x speed-up for Conway's Game of Life. **Right:** Training speed comparison between CAX and the official TensorFlow implementation for neural cellular automata experiments. CAX achieves a 1.5x speed-up on the Self-classifying MNIST Digits task.

CAX leverages JAX's powerful vectorization and scan capabilities to achieve remarkable speed improvements over existing implementations. Our benchmarks, conducted on a single NVIDIA RTX A6000 GPU, demonstrate significant performance gains across various cellular automata models. For Elementary Cellular Automata, CAX achieves a 1,400x speed-up compared to CellPyLib. In simulations of Conway's Game of Life, a 2,000x speed-up is observed relative to CellPyLib.

Furthermore, in the domain of Neural Cellular Automata, specifically the Self-classifying MNIST Digits experiment, CAX demonstrates a 1.5x speed-up over the official TensorFlow implementation. These performance improvements, illustrated in Figure 3, are made possible by JAX's efficient vectorization and the use of its scan operation for iterative computations. The following code snippet exemplifies how CAX utilizes JAX's scan function to optimize multiple CA steps:

```python
def step(carry: tuple[CA, State], input: Input | None) -> tuple[tuple
    [CA, State], State]:
    ca, state = carry
    state = ca.step(state, input)
    return (ca, state), state if all_steps else None

(_, state), states = nnx.scan(
    step,
    in_axes=(nnx.Carry, input_in_axis),
    length=num_steps,
)((self, state), input)
```

This optimized approach allows for rapid execution of complex CA simulations, opening new possibilities for large-scale experiments and real-time applications.

### 3.2.2 UTILITIES

CAX offers a rich set of utility functions to support various aspects of cellular automata research. A high-quality implementation of the sampling pool technique is provided, which is crucial for training stable growing neural cellular automata (Mordvintsev et al., 2020). To facilitate the training of unsupervised neural cellular automata and enable generative modeling within the CA framework, CAX incorporates a variational autoencoder implementation. Additionally, the library provides utilities for handling image and emoji inputs, allowing for diverse and visually engaging CA experiments. These utilities are designed to streamline common tasks in CA research, allowing researchers to focus on their specific experiments rather than reimplementing standard components.

### 3.2.3 DOCUMENTATION AND EXAMPLES

CAX prioritizes user experience and ease of adoption through comprehensive documentation and examples. The entire library is thoroughly documented, with typed classes and functions accompanied by descriptive docstrings. This ensures users have access to detailed information about CAX's functionality and promotes clear, type-safe code. To help users get started and showcase advanced usage, CAX offers a collection of tutorial-style interactive Colab notebooks. These notebooks demonstrate various applications of the library and can be run directly in a web browser without any prior setup, making it easy for new users to explore CAX's capabilities.

For easy access and integration into existing projects, CAX can be installed directly via PyPI, allowing users to quickly incorporate it into their Python environments. The library maintains high standards of code quality, with extensive unit tests covering a significant portion of the codebase. Continuous Integration (CI) pipelines ensure that all code changes are thoroughly tested and linted before integration. These features collectively make CAX not just a powerful tool for cellular automata research, but also an accessible and user-friendly library suitable for both novice and experienced researchers in the field.

## 4 IMPLEMENTED CELLULAR AUTOMATA AND EXPERIMENTS

To showcase the versatility and capabilities of the library, we show that CAX supports a wide array of cellular automata, ranging from classical discrete models to advanced continuous CAs and in-

cluding neural implementations. In this section, we provide an overview of these implementations, demonstrating the library's flexibility in handling various dimensions and types (Table 1).

We begin with three classic models that highlight CAX's ability to support both discrete and continuous systems across different dimensions. The Elementary CA, a foundational one-dimensional discrete model studied extensively by Wolfram (2002), demonstrates CAX's efficiency in handling simple discrete systems. Conway's Game of Life (Gardner, 1970), a well-known two-dimensional model, showcases CAX's capability in simulating complex emergent behaviors in discrete space. Lenia (Chan, 2019), a continuous, multi-dimensional model, illustrates CAX's flexibility in supporting more complex, continuous systems in arbitrary dimensions.

Furthermore, we have replicated four prominent NCA experiments that have gained significant attention in the field. The Growing NCA (Mordvintsev et al., 2020) demonstrates CAX's ability to handle complex growing patterns and showcases the implementation of the sampling pool technique, crucial for stable growth and regeneration. The Growing Conditional NCA (Sudhakaran et al., 2022) utilizes CAX's Controllable CA capabilities, as introduced in Section 2.2 allowing for targeted pattern generation. The Growing Unsupervised NCA (Palm et al., 2021) highlights CAX's versatility in incorporating advanced machine learning techniques, specifically the use of a Variational Autoencoder within the NCA framework. The Self-classifying MNIST Digits (Randazzo et al., 2020) showcases CAX's capacity for self-organizing systems with global coordination via local interactions, contrasting with growth-based tasks.

These implementations not only validate CAX's performance and flexibility but also serve as valuable resources for researchers looking to build upon or extend these models. We complement these implementations with three novel experiments, which will be detailed in the following section.

Table 1: Overview of Cellular Automata implemented in CAX

| Cellular Automata | Reference | Type | Dimensions |
|---|---|---|---|
| Elementary Cellular Automata | Wolfram (2002) | Discrete | 1D |
| Conway's Game of Life | Gardner (1970) | Discrete | 2D |
| Lenia | Chan (2019) | Continuous | ND |
| Growing Neural Cellular Automata | Mordvintsev et al. (2020) | Neural | 2D |
| Growing Conditional Neural Cellular Automata | Sudhakaran et al. (2022) | Neural | 2D |
| Growing Unsupervised Neural Cellular Automata | Palm et al. (2021) | Neural | 2D |
| Self-classifying MNIST Digits | Randazzo et al. (2020) | Neural | 2D |
| Diffusing Neural Cellular Automata | Section 5.1 | Neural | 2D |
| Self-autoencoding MNIST Digits | Section 5.2 | Neural | 3D |
| 1D-ARC Neural Cellular Automata | Section 5.3 | Neural | 1D |

# 5 NOVEL NEURAL CELLULAR AUTOMATA EXPERIMENTS

## 5.1 DIFFUSING NEURAL CELLULAR AUTOMATA

In this experiment, we introduce a novel training procedure for NCA, inspired by diffusion models. Traditionally, NCAs have predominantly relied on growth-based training paradigms, where the state is initialized with a single alive cell and trained to grow towards a target pattern (Sudhakaran et al., 2022; Mordvintsev et al., 2020; Palm et al., 2021). However, this approach often faces challenges in maintaining stability and achieving consistent results (Mordvintsev et al., 2020). The conventional

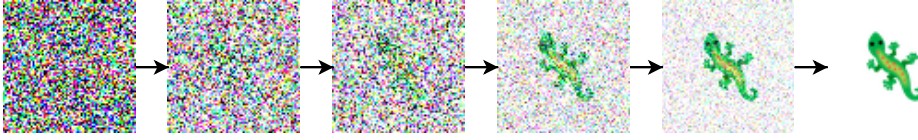

Figure 4: Inspired by diffusion models, the NCA learns to denoise images over a fixed number of steps. The process evolves from pure noise (left) to a target pattern (right).

NCA training method typically employs a "sample pool" strategy to address stability issues and encourage the formation of attractors. This approach involves maintaining a diverse pool of intermediate states, sampling from this pool for training, and periodically updating it with newly generated states. By exposing the NCA to various intermediate configurations and consistently guiding them towards the target pattern, the sample pool method helps shape the system's dynamics, making the desired pattern a more robust attractor in the state space.

Our proposed diffusion-inspired approach offers several advantages over the traditional growing mechanism. First, unlike the growing mechanism, our diffusion-based approach does not require a sample pool, which simplifies the training process and reduces memory requirements, making it more efficient and scalable. Second, our diffusion-inspired approach naturally guides the NCA towards more stable dynamics, effectively creating a stronger attractor basin around the target pattern. In Figure 5, we compare the regeneration capabilities of growing NCAs with diffusing NCAs. We create an artificial damage by cutting the tail of the gecko and observe that diffusing NCA demonstrate emergent regenerating capabilities.

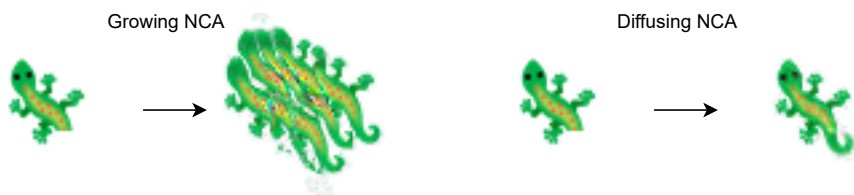

Figure 5: Diffusing NCAs demonstrate emergent regenerating capabilities compared to growing NCAs that are unstable if not trained explicitly to regenerate and recover from damage.

## 5.2 SELF-AUTOENCODING MNIST DIGITS

In this experiment, we draw inspiration from Randazzo et al. (2020) where an NCA is trained to classify MNIST digits through local interactions. In their work, each cell (pixel) of an MNIST digit learns to output the correct digit label through local communication with neighboring cells. The NCA demonstrates the ability to reach global consensus on digit classification, maintain this classification over time, and adapt to perturbations or mutations of the digit shape. Their model showcases emergent behavior, where simple local rules lead to complex global patterns, analogous to biological systems achieving anatomical homeostasis.

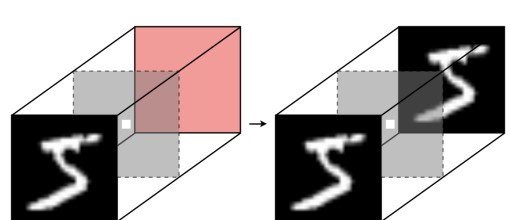

Figure 6: The 3D NCA is initialized with an MNIST digit (left). The NCA learns to reconstruct the digit on the opposite red face (right).

Building upon this concept, we propose a novel experiment that could be termed "Self-autoencoding MNIST Digits". In this setup, we utilize a three-dimensional NCA initialized with an MNIST digit on one face, see Figure 6. The objective of the NCA is to learn a rule that will replicate the MNIST digit on its opposite face (red face). However, we introduce a critical constraint: in the middle of the NCA, there is a mask where cells cannot be updated, effectively preventing direct communication between the two faces. Crucially, we allow for a single-cell wide hole in the center of this mask, creating a minimal channel for information transfer.

To successfully replicate the MNIST digit on the opposite face, the NCA must develop a sophisticated rule set that accomplishes two key tasks. First, it must encode the MNIST image into a compressed form that can pass through the single-cell hole. Second, it must then decode this information on the other side to accurately reconstruct the original digit. A notable aspect of this result is that each cell in the NCA performs an identical local update rule, contributing to the system's overall emergent behavior. As shown in Figure 7, the NCA successfully reconstructs MNIST digits on

the red face, demonstrating its ability to encode, transmit, and decode complex visual information through a minimal channel. This experiment highlights the power of NCAs in learning complex information processing tasks using simple, uniform rules, while demonstrating CAX's ability to support sophisticated 3-dimensional CA rules.

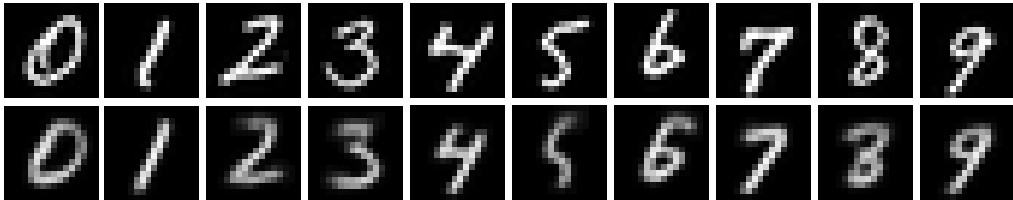

Figure 7: The top row shows the original digits from the test set, while the bottom row displays the corresponding reconstructions on the red face of the NCA.

## 5.3 1D-ARC NEURAL CELLULAR AUTOMATA

In this experiment, we train a one-dimensional NCA on the 1D-ARC dataset (Xu et al., 2024). The 1D-ARC dataset is a novel adaptation of the original Abstraction and Reasoning Corpus (Chollet, 2019) (ARC), designed to simplify and streamline research in artificial intelligence and language models. By reducing the dimensionality of input and output images to a single row of pixels, 1D-ARC maintains the core knowledge priors of ARC while significantly reducing task complexity. For example, the tasks in 1D-ARC include "Static movement by 3 pixels", "Fill", and "Recolor by Size Comparison". For a full description of the dataset, see the project page. Our experiment focuses on training an NCA to solve the 1D-ARC tasks. Each input sample consists of a single row of colored pixels and a corresponding target row. The NCA's objective is to transform the input into the target through successive applications of its rule. We consider a task successful if all pixels in the NCA's output match the target pixels after a predetermined fixed number of steps.

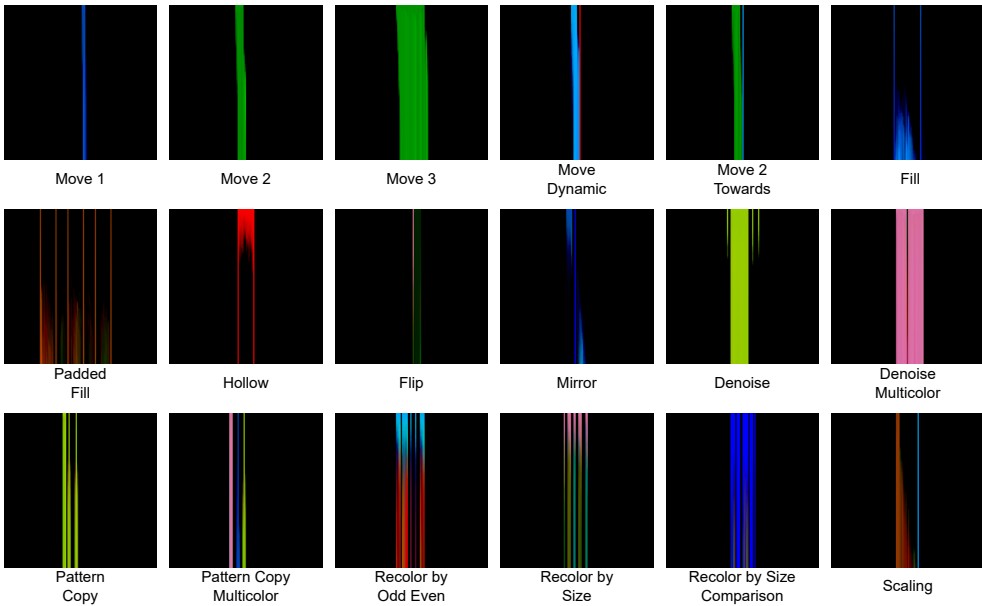

Figure 8: 1D-ARC NCA space-time diagrams for each task. The top row of pixels in each image is the input. Subsequent rows of pixels show the NCA's intermediate steps as it attempts to transform the input into the target. The bottom row of pixels represents the NCA's final output after a fixed number of steps, which is compared to the target for task completion.

The primary goal of this experiment is for the NCA to learn a generalizable rule from the training set, enabling it to solve unseen examples in the test sets. This challenge tests the NCA's ability to infer abstract patterns and apply them to new situations, a key aspect of human-like reasoning. Figure 8 illustrates the NCA's "reasoning" on all 1D-ARC tasks. The visualization shows the input at the top, intermediate steps, and final output of the NCA at the bottom of each image, and is called a space-time diagram.

To evaluate the NCA's performance, we compare it to GPT-4, a state-of-the-art language model, on the 1D-ARC test set. Table 2 presents the accuracy of the NCA and GPT-4 across 18 different task types. The GPT-4 values are direct-grid results, directly taken from Xu et al. (2024). Notably, the NCA outperforms GPT-4 on several tasks, particularly those involving movement, pattern copying, and denoising. Overall, the NCA achieves a total accuracy of 60.12% compared to GPT-4's 41.56%, as reported by Xu et al. (2024).

These results demonstrate the potential of NCAs in solving abstract reasoning tasks, even outperforming sophisticated language models in certain domains. The NCA's success in tasks like "Move 3" and "Pattern Copy Multicolor" showcases its ability to learn complex spatial transformations and apply them consistently.

Table 2: GPT-4 and NCA accuracy in percentage on all tasks from the 1D-ARC test set. The GPT-4 values are direct-grid approach, directly taken from Xu et al. (2024).

| Task | GPT-4 | NCA |
|---|---|---|
| Move 1 | 66 | **100** |
| Move 2 | 26 | **100** |
| Move 3 | 24 | **100** |
| Move Dynamic | **22** | 12 |
| Move 2 Towards | 34 | **98** |
| Fill | **66** | **66** |
| Padded Fill | 26 | **28** |
| Hollow | 56 | **98** |
| Flip | **70** | 28 |
| Mirror | **20** | 6 |
| Denoise | 36 | **100** |
| Denoise Multicolor | **60** | 58 |
| Pattern Copy | 36 | **100** |
| Pattern Copy Multicolor | 38 | **100** |
| Recolor by Odd Even | **32** | 0 |
| Recolor by Size | **28** | 0 |
| Recolor by Size Comparison | **20** | 0 |
| Scaling | **88** | **88** |
| Total | 41.56 | **60.12** |

However, the NCA struggles with tasks involving more abstract concepts like odd-even distinctions or size comparisons. This limitation suggests areas for future improvement, possibly through the integration of additional priors or more sophisticated architectures. While the average of NCA outperforms GPT4, it is interesting to note that GPT4 performs equally in every task, while NCA completely fails on some of them (0% accuracy). This opens interesting questions for future work. This experiment not only highlights the capabilities of NCAs in abstract reasoning tasks but also demonstrates CAX's flexibility in implementing and training NCA models for diverse applications.

## 6 CONCLUSION

In this paper, we introduce CAX: Cellular Automata Accelerated in JAX, an open-source library, designed to provide a high-performance and flexible framework to accelerate cellular automata research. CAX provides substantial speed improvements over existing implementations, enabling researchers to run complex simulations and experiments more efficiently.

CAX's flexible architecture supports a wide range of cellular automata types across multiple dimensions, from classic discrete models to advanced continuous and neural variants. Its modular design, based on customizable perceive and update components, facilitates rapid experimentation and development of novel CA models, enabling efficient exploration of new ideas.

CAX's comprehensive documentation, example notebooks, and seamless integration with machine learning workflows not only lower the barrier to entry but also promote reproducibility and collaboration in cellular automata research. We hope this accessibility will accelerate the pace of discovery by attracting new researchers.

In the future, we envision several exciting directions, such as expanding the model zoo to implement and optimize a wider range of cellular automata models, and exploring synergies between cellular automata and other approaches, such as reinforcement learning or evolutionary algorithms.

## ACKNOWLEDGMENTS

We thank Gabriel Béna and Arthur Braida for insightful discussions and feedback. We also acknowledge the financial support provided by a grant from G-Research.

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

## A    HYPERPARAMETERS

This appendix provides detailed hyperparameters for the three novel neural cellular automata (NCA) experiments introduced in Section 5.  These hyperparameters govern the architecture, training process, and simulation dynamics of each experiment.  For further details on the experimental setup, refer to the respective subsections in Section 5 or to the notebooks on https://github.com/maxencefaldor/cax.

Table 3: Hyperparameters for Diffusing Neural Cellular Automata (see Section 5.1)

| Parameter | Value |
|---|---|
| Spatial dimensions | $(72, 72)$ |
| Channel size | 64 |
| Number of kernels | 3 |
| Hidden size | 256 |
| Cell dropout rate | 0.5 |
| Batch size | 8 |
| Number of steps | 64 |
| Learning rate | 0.001 |

Table 4: Hyperparameters for Self-autoencoding MNIST Digits (see Section 5.2)

| Parameter | Value |
|---|---|
| Spatial dimensions | $(16, 16, 32)$ |
| Channel size | 32 |
| Number of kernels | 4 |
| Hidden size | 256 |
| Cell dropout rate | 0.5 |
| Batch size | 8 |
| Number of steps | 96 |
| Learning rate | 0.001 |
| Pool size | 1,024 |

Table 5: Hyperparameters for 1D-ARC Neural Cellular Automata (see Section 5.3)

| Parameter | Value |
|---|---|
| Spatial dimensions | $(96)$ |
| Channel size | 64 |
| Number of kernels | 2 |
| Hidden size | 256 |
| Cell dropout rate | 0.5 |
| Batch size | 8 |
| Number of steps | 64 |
| Learning rate | 0.001 |

