# OpenReview forum: "CAX: Cellular Automata Accelerated in JAX"
_ICLR.cc/2025/Conference — ICLR 2025 Oral_

### Official Review · Reviewer_XZsB · 2024-11-01

**Soundness:** 4
**Presentation:** 3
**Contribution:** 4
**Rating:** 8
**Confidence:** 5

**Summary:**

The authors have created an open-sourced python library called CAX (cellular automata accelerated in JAX) that implements a number of recent developments/experiments in the field of cellular automata as JAX models which allows for highly efficient and parallel computing. The authors also implement tools to customize these implementation for future researchers to run their own experiments in order to have a unified library to be used for efficient CA computations. Furthermore, they implement 3 new experiments with this library to demonstrate these customization tools (a diffusion CA model, an auto-encoding CA model, and neural CA model trained on 1D-ARC tasks).

**Strengths:**

Originality: While this paper does not necessarliy re-invent new methods, they have created an original and unified tool to be used by practitioners of cellular automata in a highly efficient way for a broad set of possible implementations. This is a tool that will surely accelerate the study of emergence and self-organization in cellular automata.

Quality/Clarity: The paper is clearly written with sufficient background context to motivate the necessity of this tool as well as citing the relevant literature that has sparked recent interest in deeper neural cellular automata architectures.

Significance: Related to the originality of this tool, it is significant for practitioners of this field and lowers the barrier of entry for researchers to produce highly efficient code and run larger scale simulations and experiments. As the field of cellular automata is deeply intertwined with the study of emergence, scaling such simulations is a crucial requirement to push the field to new grounds. With such a tool, such experiments become much more feasible and cheaper for researchers.

**Weaknesses:**

- the performance increases plotted in figure 3 I feel are not sufficiently demonstrating the utility of such a library. I say this as a user of JAX and PyTorch and know that there are many performance increases when using JAX, and so there needs to be a much more comprehensive case being made here that JAX is actually introducing large speed ups. This is also increasingly important as there doesn't seem to be a massive speed up in the "Growing cellular automata" experiment, so perhaps the authors should compute performance changes for a larger set of experiments. (For example for the [self-organizing textures](https://distill.pub/selforg/2021/textures/) experiment, or [cellular automata controlling cartpole](https://arxiv.org/abs/2106.15240) experiment.

- as someone that has had to write JAX code for my own custom CA implementations, it would be great if the appendix included some annotated code that explains how efficient JAX implementations can be written. I know that curious readers can always read the code on github, and there are some examples shown such as on line 281-289 as well as the example notebook in the appendix (but it has minimal comments), but it would be extremely helpful for new programmers to understand the thought process for writing such a library. This is doubly important because when trying to implement this library for custom experiments, a certain degree of programming in JAX will be required. This is possibly slightly outside the scope or responsibility of the authors, but I think it would go a long way for allowing researchers to adopt this library if there were tutorials or detailed annotations of code that explain this process (I know for myself this would have been very helpful a few years ago).

- I very quickly tried to run one of the colab notebooks for the 1D-ARC tasks and had problems with importing. I did not test with all the notebooks but this should probably be double-checked across the board. I recieved two errors:
   - (on the second cell: `%pip install "cax[examples] @ git+https://github.com/879f4cf7/cax.git"`): after installing most packages, I recieved the following message: `ERROR: pip's dependency resolver does not currently take into account all the packages that are installed. This behaviour is the source of the following dependency conflicts.
gcsfs 2024.10.0 requires fsspec==2024.10.0, but you have fsspec 2024.9.0 which is incompatible.`
    - (on the third cell: `from cax.core.update.residual_update import ResidualUpdate`):
~~~
---------------------------------------------------------------------------
ModuleNotFoundError                       Traceback (most recent call last)
<ipython-input-3-7bc79b5c20c3> in <cell line: 11>()
      9 from cax.core.perceive.depthwise_conv_perceive import DepthwiseConvPerceive
     10 from cax.core.perceive.kernels import grad_kernel, identity_kernel
---> 11 from cax.core.update.residual_update import ResidualUpdate
     12 from flax import nnx
     13 from tqdm.auto import tqdm

1 frames
/usr/local/lib/python3.10/dist-packages/cax/core/update/mlp_update.py in <module>
      5 import jax.numpy as jnp
      6 from flax import nnx
----> 7 from flax.nnx.nnx.nn import initializers
      8 from flax.nnx.nnx.nn.linear import default_kernel_init
      9

ModuleNotFoundError: No module named 'flax.nnx.nnx'
~~~

**Questions:**

I don't have any major questions regarding this package other than the reason for the import crashes in the notebook I listed in the weaknesses.

---

> ### Author Response · Authors · 2024-11-21
>
> Thank you for your constructive feedback. We appreciate the opportunity to address your concerns.
>
> ---
>
> > as someone that has had to write JAX code for my own custom CA implementations, it would be great if the appendix included some annotated code that explains how efficient JAX implementations can be written. I know that curious readers can always read the code on github, and there are some examples shown such as on line 281-289 as well as the example notebook in the appendix (but it has minimal comments), but it would be extremely helpful for new programmers to understand the thought process for writing such a library. This is doubly important because when trying to implement this library for custom experiments, a certain degree of programming in JAX will be required.
>
> Thank you for this insightful suggestion about providing more detailed code explanations. To address this need and help researchers effectively use and contribute to CAX, we have added a comprehensive [CONTRIBUTING.md](https://github.com/879f4cf7/cax/blob/main/CONTRIBUTING.md) file to the repository. This guide includes not only standard contribution guidelines but also a dedicated section on CAX architecture design principles.
>
> Beyond the usual content about bug reports, pull requests, and coding style, we've added a "Designing Efficient CAX Architectures" section that explains:
> - How to structure CA models using perceive and update modules
> - Best practices for leveraging JAX/Flax functionality
> - An example structure
> - A common training loop for training neural cellular automata
> - Common patterns and pitfalls when implementing custom CAs
>
> This documentation should help both new users getting started with CAX and experienced developers looking to implement custom experiments efficiently.
>
>
> > the performance increases plotted in figure 3 I feel are not sufficiently demonstrating the utility of such a library. I say this as a user of JAX and PyTorch and know that there are many performance increases when using JAX, and so there needs to be a much more comprehensive case being made here that JAX is actually introducing large speed ups. This is also increasingly important as there doesn't seem to be a massive speed up in the "Growing cellular automata" experiment, so perhaps the authors should compute performance changes for a larger set of experiments. (For example for the self-organizing textures experiment, or cellular automata controlling cartpole experiment.
>
> Thank you for this feedback about performance analysis. In CAX, cellular automata models inherit from Flax's `nnx.Module` and use standard components like convolution layers, linear layers, and dropout mechanisms. Once defined, these models can be trained like any other recurrent architecture. As such, CAX naturally inherits the performance characteristics and scaling properties of the JAX/Flax ecosystem, which are already well-documented. A CA model in CAX is defined through perceive and update modules. The overall performance depends on the efficiency of these modules - when they are well-implemented using optimized Flax components, the resulting CA will be efficient.
>
> Additionally, since CAX is the first hardware-accelerated library for cellular automata, running more extensive benchmarks or scaling studies would provide limited insight due to the absence of comparable baselines. The value of CAX lies not just in its performance but in providing a structured, modular framework for implementing cellular automata with hardware acceleration.
>
> However, we agree with the reviewer that we could compute performance changes for a larger set of experiments. Our vision for CAX includes expanding its scope to encompass a broader range of experiments, including the notable self-organizing textures implementation. While we aim to add the self-organizing textures comparison before the end of the rebuttal period, we want to be transparent that, given our concurrent commitments as authors and reviewers of multiple papers, we cannot guarantee its completion within this limited timeframe.
>
>
> > I very quickly tried to run one of the colab notebooks for the 1D-ARC tasks and had problems with importing. I did not test with all the notebooks but this should probably be double-checked across the board. I recieved two errors.
>
> Thank you for reporting these installation issues. We apologize for the difficulties you encountered running our notebooks.
>
> The first error regarding `fsspec` versions can be safely ignored.
>
> The second error occurred due to an API change in Flax between versions 0.9.0 and 0.10.0 that happened between the submission and the rebuttal period.
>
> We have updated the anonymous repository to be compatible with the latest Flax version, and the notebook is now running correctly. Please, let us know if you still have any issues.
>
> ---
>
> We believe these responses address your concerns. We welcome any further questions or clarifications you may need.

---

> > ### Comment · Reviewer_XZsB · 2024-11-26
> >
> > Thank you to the authors for responding to my questions/comments. I won't be changing my scores as they were quite positive to start, and I do believe this will be a very relevant contribution to the field, though I still believe there are some iterations left to make the library as accessible as it has the potential to be. The contributions.md file certainly is a good step in the right direction, allowing people to contribute to the work and be a community accessible open-sourced library.
> >
> > I would re-iterate some of the comments made by other reviewers calling for more hard/soft-scaling tests as well as further expansion of documentation and annotated examples, though I also understand the time constraints for the authors for this specific deadline may not allow for everything that is being asked for. I would however strongly recommend the authors to iterate on these 'community accessability' features over the next few weeks/months as this library (hopefully) grows.

---

### Official Review · Reviewer_7WC1 · 2024-11-02

**Soundness:** 3
**Presentation:** 4
**Contribution:** 2
**Rating:** 8
**Confidence:** 3

**Summary:**

The paper presents a new open-source library for performing Cellular Automata studies written in the JAX framework. The use of JAX allows for significant acceleration of such studies, and the framework reduces the amount of boilerplate code scientists need to write when starting such an investigation.

Several benchmarks are shown to demonstrate the structure of the software API and even demonstrate a result when compared with common language models.

**Strengths:**

The paper is very well written and easy to read. The authors have made presentation a clear priority in the paper and should be commended.

The software itself is also a great idea. Reducing the time it takes for people to start working on these kinds of studies and to do so in an accelerated, modern way is very useful and this work presents a framework doing exactly that.

**Weaknesses:**

While the paper presents a valuable tool, it does so in a very surface-level way. Most software discussions are in the API demonstration, which shows users how to put it into practice. This is important, but in an academic paper presenting the framework, the framework should take centre stage. Further, benchmarks against common test systems are very welcome in a software paper, but the comparison with GPT-4, while certainly interesting, is more of a research result in and of itself and not directly related to the software. I am sure there is more to the code that replacing numpy with JAX, but that should be laid out and explained in the manuscript.

My biggest concern is that ICLR isn't quite the correct platform for such a paper. I understand the desire for it to go there as it is related to ML but it is, at the end of the day, software. If the paper were to present a novel approach to structuring CA algorithms and packaging that up, it would be a different story. However, this decision is likely up to conference organizers. I can see the benefit of using ICLR as a platform to spread this software and perhaps help a lot of scientists. But from a newness perspective, it is not a breakthrough.

**Questions:**

* Why not run proper hard and soft scaling tests in the performance comparison? It is always hard to show few numbers in a speed comparison.
* Further, is the performance improvement only due to JAX or have you also written the code in a way that further improves parts of the algorithms? If so, that would make a key result in the paper and improve its novelty.

---

> ### Author Response · Authors · 2024-11-21
>
> Thank you for your thoughtful feedback and the opportunity to clarify these points.
>
> ---
>
> > Further, benchmarks against common test systems are very welcome in a software paper, but the comparison with GPT-4, while certainly interesting, is more of a research result in and of itself and not directly related to the software.
>
> Thank you for this comment. We agree that the GPT-4 comparison is more of a research result and not directly related to the software evaluation. As stated in our abstract and introduction, the experiments in Section 6 were designed to demonstrate the library's capabilities, not as full-fledged research contributions. Given that the paper's primary contribution is the library itself, we prioritized explaining its architecture and design principles.
>
> Regarding benchmarks, we provide comparisons against CellPyLib on two canonical test systems: Conway's Game of Life and Elementary Cellular Automata. While these show significant speed-ups (2000x), we acknowledge that such comparisons have limited value since CellPyLib serves different use cases and isn't hardware-accelerated. We include these mainly to give CA practitioners an indication of potential performance gains when moving to hardware-accelerated implementations.
>
> For Neural CA experiments, we could only compare against the original papers' implementations since no unified NCA library exists. While more extensive benchmarking would be valuable, the absence of other hardware-accelerated CA libraries makes this impossible at present. We have expanded our discussion of the library's engineering aspects, particularly how we leverage Flax's nnx.Module for hardware-accelerated implementations of CA components.
>
> For NCA experiments, we would have liked to provide a more thorough benchmark and comparison against a more extensive range of library. However, at the moment, to the best of our knowledge, no NCA library exists so we could only compare against fragmented official implementations on two NCA experiments coming from the original papers.
>
>
> > My biggest concern is that ICLR isn't quite the correct platform for such a paper. If the paper were to present a novel approach to structuring CA algorithms and packaging that up, it would be a different story.
>
> First, we would like to note that "Software Libraries" is explicitly listed in ICLR's official Call for Papers, and the conference specifically welcomes papers exploring implementation issues, parallelization, software platforms, and hardware. The present work falls primarily under the area of infrastructure, software libraries, hardware, and systems.
>
> Moreover, NCAs represent a fundamental approach to machine learning where complex behaviors emerge from learning simple, local rules and representations. This aligns directly with ICLR's core focus on representation learning - instead of learning global solutions, NCAs learn local update rules that, when applied iteratively, lead to emergent problem-solving capabilities. As a machine learning architecture, NCAs have demonstrated state-of-the-art performance in various domains, such as dynamic texture synthesis, and ViTCA architectures have shown superior performance across all benchmarks and nearly every evaluation metric on denoising autoencoding tasks.
>
> While CAX's main contribution is a general library to implement various CA methods, the key way we enable this is precisely by introducing a novel approach to structuring CA algorithms through our controllable CA framework, as illustrated in Figure 3 of our paper.
>
> Specifically, we introduce a fundamental architectural division between Perceive and Update modules. This separation enables strong modularity, component reuse, and clear division of responsibilities, as detailed in Section 3. The Perceive module handles neighborhood observations while the Update module determines state transitions, creating a flexible yet powerful structure that can support diverse CA types across multiple dimensions.
>
> This architectural contribution represents a novel framework for organizing CA algorithms, making them more maintainable, reusable, and easier to implement while maintaining high performance through hardware acceleration. The table below demonstrates how this architecture effectively unifies different CA methods under a common structure:
>
> | Model | Perceive | Update |
> |--------|----------|---------|
> | elementary_ca | MoorePerceive | ElementaryUpdate |
> | life | MoorePerceive | LifeUpdate |
> | lenia | LeniaPerceive | LeniaUpdate |
> | growing_nca | ConvPerceive | NCAUpdate |
> | growing_conditional_nca | ConvPerceive | NCAUpdate |
> | growing_unsupervised_nca | ConvPerceive | NCAUpdate |
> | self_classifying_mnist | ConvPerceive | NCAUpdate |
> | self_autoencoding_mnist | ConvPerceive | NCAUpdate |
> | diffusing_nca | ConvPerceive | NCAUpdate |
> | 1d_arc_nca | ConvPerceive | ResidualUpdate |

---

> ### Author Response · Authors · 2024-11-21
>
> > I can see the benefit of using ICLR as a platform to spread this software and perhaps help a lot of scientists.
>
> Thank you for this comment. Indeed, we believe that presenting CAX at ICLR would benefit the research community. As the first hardware-accelerated cellular automata library, CAX represents a significant technical advance that makes sophisticated CA experiments accessible to researchers. Its publication at ICLR would help disseminate these capabilities widely and inspire future innovations in the field.
>
> > While the paper presents a valuable tool, it does so in a very surface-level way. Most software discussions are in the API demonstration, which shows users how to put it into practice. This is important, but in an academic paper presenting the framework, the framework should take centre stage.
>
> Thank you for your valuable feedback. In line with your remark, we have updated the manuscript to put the framework more at the center stage of the paper. In particular, we have expanded our description of the framework and its versatility, highlighting how our perceive/update architecture enables implementation of recent important NCA papers.
>
> > Further, is the performance improvement only due to JAX or have you also written the code in a way that further improves parts of the algorithms? If so, that would make a key result in the paper and improve its novelty. I am sure there is more to the code that replacing numpy with JAX, but that should be laid out and explained in the manuscript.
>
> Thank you for this feedback. We have added clarifications about these key implementation details and their contributions to performance in both the introduction and discussion sections of the paper.
>
> The performance improvements in CAX stem from both architectural design choices and extensive leveraging of JAX/Flax capabilities that are particularly well-suited for cellular automata computations.
>
> We extensively utilize JAX's vectorization capabilities through `vmap`, which is especially powerful for CA since the same rule is applied across all cells. This naturally maps to hardware acceleration patterns. Additionally, we make extensive use of Flax neural network modules (`nnx.Module`) like convolutions, MLPs, and dropout layers, which are highly optimized for modern hardware and use JAX's `scan` operator for the sequential update of the CA state.
>
> While we build upon previous work in combining CA with machine learning architectures, we also contribute novel optimized implementations. For example, we provide an efficient JAX implementation of Sample Pooling and introduce dropout-based stochastic cell updates. By leveraging existing Flax `nnx.Module` components wherever possible, we benefit from their high-quality hardware-accelerated implementations.
>
> So while JAX's capabilities are foundational to CAX's performance, our careful attention to implementation details and systematic use of optimized Flax modules further enhance the library's efficiency.
>
> ---
>
> We hope we have answered all your questions and concerns. Please do not hesitate to bring up any additional questions.

---

> > ### Comment · Reviewer_7WC1 · 2024-11-22
> > **Response to authors**
> >
> > I appreciate the authors taking the time to answer most of my concerns. I want to bring up one final concern: the problem of the performance results presented in the manuscript. It is still the correct protocol to run hard and soft scaling tests if the goal is to demonstrate the performance of the software. This will likely mirror the hard/soft scaling tests the JAX team could run, but it would still indicate any algorithmic boosts or slowdowns incurred by your implementation.
> >
> > In saying that, I would also like to apologize. I did not see the explicit call for software implementations in the call for papers. In light of this, I will withdraw my objections that the paper does not belong in ICLR and increase my score slightly.
> >
> > As a software paper, however, the focus should be on efficient, usable software rather than just demonstrating its use. This means clear architecture diagrams, algorithmic improvements, and proper scaling tests. Otherwise, it reads more like a brochure or advertisement. Figure 2 is the only figure in the main text highlighting an algorithm and is very generic. Outside this figure, we see tables of results and pictures generated from studies. The utilities section of the paper, for example, is far smaller than the examples sections, where these are the tools that are of interest. From my perspective, how the authors built the software and why this stands out are more important than demonstrating that a CA package can implement CAs.

---

### Official Review · Reviewer_BH4L · 2024-11-04

**Soundness:** 3
**Presentation:** 4
**Contribution:** 2
**Rating:** 8
**Confidence:** 3

**Summary:**

This paper presents a software library for cellular automata with GPU acceleration through the JAX library. It demonstrates this new library on MNIST digit classification and the 1D version of the Abstract Reasoning Challenge

**Strengths:**

The article is clearly written and makes a compelling argument for cellular automata. It demonstrates their utility in image analysis and even reasoning tasks. The use of GPU acceleration through JAX is timely and relevant as a new exploration of GPU acceleration capabilities. As the main contribution is a software library, it provides some experimental comparisons with other open-source libraries.

The secondary contribution of the article are three novel applications of NCA. The most concrete of these is the application to the 1D ARC dataset. This demonstrates that NCA can follow patterns from a small number of examples, allowing them to rival GPT4's in-context learning on visual reasoning tasks. To my knowledge, this is the first study of NCA for reasoning; the demonstration that NCA can do low-data pattern matching, whether or not that is "reasoning", is very interesting.

**Weaknesses:**

A question I had in reviewing this was its relevancy to ICLR. It isn't the standard ICLR paper, so I consulted ICLR's official CFP, which includes:

+ generative models
+ causal reasoning
+ infrastructure, software libraries, hardware, etc.

I believe the article fits into these categories, however the match to ICLR and the general machine learning literature could be strengthened. For example, these works use NCA in more traditional machine learning settings:

Grattarola, Daniele, Lorenzo Livi, and Cesare Alippi. "Learning graph cellular automata." Advances in Neural Information Processing Systems 34 (2021): 20983-20994.

Tesfaldet, Mattie, Derek Nowrouzezahrai, and Chris Pal. "Attention-based neural cellular automata." Advances in Neural Information Processing Systems 35 (2022): 8174-8186.

Pajouheshgar, Ehsan, et al. "Dynca: Real-time dynamic texture synthesis using neural cellular automata." Proceedings of the IEEE/CVF conference on computer vision and pattern recognition. 2023.

None of these are cited, despite being high-profile studies of NCA in machine learning (NeurIPS, CVPR). As the main contribution of this article is a software library, it would be very valuable to know if these studies or similar ones could be reproduced using CAX. For submission to ICLR, this article could do more to relate the contribution to machine learning.

The second major weakness is the limited experimental study. The studies proposed in section 5 function more as demonstrations of the CAX library than new experiments due to their lack of rigor. Only the 1D-ARC experiment presents quantified results and a comparison with another method. The novel diffusion method presented in 5.1 is only demonstrated on a single example, and the self-encoding MNIST digits experiment in 5.2 is only validated through visual comparison of 1 image per digit. As such, claims like " As shown in Figure 7, the NCA successfully reconstructs MNIST digits on the red face, demonstrating its ability to encode, transmit, and decode complex visual information
through a minimal channel" are not fully supported. Experimental details and full reporting of experimental results are necessary to support such claims. As demonstrations of the CAX library, these examples are interesting, but as experiments, they fall short.

**Questions:**

Are the mentioned works feasible or simple to implement in CAX? Specifically ViTCA is interesting and could lead to an application of NCA on the full ARC set, but is the attention mechanism possible to implement in CAX?

How does the damage repair example of Figure 5 incorporate diffusion? Do you renoise the "damaged" image or is it simply through training on the diffusion process that it is able to recreate the original image? Could this process be used to train a generative NCA over a distribution of images instead of a single one?

---

> ### Author Response · Authors · 2024-11-21
>
> Thank you for your thorough review. We appreciate the opportunity to address your concerns.
>
> ---
>
> > I believe the article fits into these categories, however the match to ICLR and the general machine learning literature could be strengthened.
>
> Thank you for highlighting these important references. We have incorporated the suggested citations and expanded our discussion to better position CAX within the ML literature.
>
> NCAs represent a fundamental approach to machine learning where complex behaviors emerge from learning simple, local rules and representations. This aligns directly with ICLR's core focus on representation learning - instead of learning global solutions, NCAs learn local update rules that, when applied iteratively, lead to emergent problem-solving capabilities.
>
> Your feedback has helped us better articulate these connections and strengthen the paper's relevance to the ICLR community.
>
>
> > Are the mentioned works feasible or simple to implement in CAX?
>
> We appreciate your interest in implementing advanced CA architectures. Yes, all the mentioned works are feasible and should be straightforward to implement in CAX. We have added an explanation in the manuscript saying that these papers can easily be implemented with CAX.
>
> Since all CA in CAX inherit from the Flax's `nnx.Module`, implementing attention mechanisms or any other modern ML architecture is fully supported. In fact, any architecture that can be implemented in Flax can be naturally integrated into CAX. This includes attention mechanisms, graph neural networks, and other sophisticated neural architectures from the ML literature.
>
> Our vision for the library is to serve as a comprehensive platform that continuously grows with more CA architectures and experiments, spanning from classical models to cutting-edge neural approaches. We are actively expanding our collection of implementations and welcome contributions from the community.
>
>
> > Specifically ViTCA is interesting and could lead to an application of NCA on the full ARC set, but is the attention mechanism possible to implement in CAX?
>
> Thank you for the comment regarding ViTCA. Indeed, ViTCA is a very interesting approach and we confirm that the attention mechanism is fully implementable within the CAX framework.
>
> We have implemented and added a proof of concept for ViTCA in the anonymous repository. We encourage the reviewers to explore the example Colab [notebook](https://colab.research.google.com/github/879f4cf7/cax/blob/main/examples/vitca.ipynb) that is now available in the [README](https://github.com/879f4cf7/cax).
>
> In CAX, we implemented ViTCA through a new perceive module called `ViTPerceive`, available in [cax/core/perceive/vit_perceive.py](https://github.com/879f4cf7/cax/blob/main/cax/core/perceive/vit_perceive.py). The `ViTPerceive` class processes cells by feeding them into a multi-head self-attention mechanism with localized attention, specifically using the Moore neighborhood.
>
> While this implementation serves as a proof of concept and doesn't implement all details from the original paper, it successfully demonstrates that ViTCA can be integrated into CAX. For transparency, our current implementation has two notable simplifications compared to the original paper:
>
> - We do not currently use layer normalization
> - We assume a one-to-one mapping between pixels and patches
>
> These simplifications allow us to demonstrate the core concepts while maintaining clarity in the implementation. The proof of concept successfully shows that the attention mechanism can be effectively implemented within CAX's architectural constraints.

---

> ### Author Response · Authors · 2024-11-21
>
> > The studies proposed in section 5 function more as demonstrations of the CAX library than new experiments...
>
> Thank you for this comment. As stated in both our abstract and introduction, these experiments were specifically designed to demonstrate and showcase the library's capabilities: "_We demonstrate CAX's performance and flexibility through a wide range of benchmarks and applications_" and "_Furthermore, we demonstrate CAX's potential to accelerate research by presenting a collection of three novel cellular automata experiments, each implemented in just a few lines of code thanks to the library's modular architecture_".
>
> We acknowledge that these experiments are not intended as full-fledged research contributions, and we would be happy to revise any language in the manuscript that might suggest otherwise.
>
> We faced a fundamental trade-off in the paper's focus and limited space. A deeper experimental analysis would have come at the expense of thoroughly presenting the library's core innovations - the controllable CA framework and the perception/update design pattern. Each concept discussed in Section 6 could merit its own paper. Given that this paper's primary contribution is the library itself rather than the experiments, we chose to prioritize explaining the framework's architecture and design principles.
>
> While the experimental validation could be more extensive, we prioritized transparency and reproducibility. All experiments are available as notebook examples that can be run in minutes using Colab. We believe this practical accessibility, allowing readers to directly verify and build upon our work, provides significant value - potentially more than extensive experimental results without accessible code or straightforward replication paths.
>
> We are delighted that the reviewers find these experiments interesting, and we hope they can inspire and stimulate further research in cellular automata. We look forward to seeing how the community builds upon these initial experiments to develop them into full research contributions.
>
>
> > How does the damage repair example of Figure 5 incorporate diffusion? Do you renoise the "damaged" image or is it simply through training on the diffusion process that it is able to recreate the original image?
>
> Thank you for this question about the damage repair example. The reconstruction occurs simply by applying the learned CA rule to the damaged image, without any renoising step. Noise is only added during training.
>
> The key insight is that the diffusion-based training procedure naturally endows the NCA with robust regenerative capabilities. By learning to denoise images at various noise levels during training, the CA develops dynamics with strong attractor basins around target patterns. When damage occurs, these learned dynamics guide the system back toward the original pattern.
>
> This emergent regenerative behavior is a direct consequence of the diffusion training process - we do not explicitly train for damage repair. Rather, the CA learns stable dynamics that can both denoise images and repair damage, demonstrating how diffusion training leads to robust pattern formation capabilities.
>
>
> > Could this process be used to train a generative NCA over a distribution of images instead of a single one?
>
> Excellent question. While the training procedure could theoretically be extended to handle a distribution of images rather than a single target, determining whether it would yield good results remains an open question. This is an exciting direction for future research.
>
> ---
>
> We hope we have answered all your questions and concerns. Please let us know if you have any additional questions or concerns.

---

> > ### Comment · Reviewer_BH4L · 2024-11-22
> > **Increasing score**
> >
> > Thank you for your comprehensive response. I believe that you have addressed the concerns that I raised, and the quick implementation of ViTCA into CAX is an impressive demonstration of the utility of this library. I will raise my review score to an 8.

---

### Official Review · Reviewer_5JEh · 2024-11-04

**Soundness:** 2
**Presentation:** 3
**Contribution:** 3
**Rating:** 8
**Confidence:** 4

**Summary:**

The paper present a new library that can model several types of cellular automata (CAs), ranging from simply Wolfram-style automata and Conway's game of life to continuous and/or complex neural cellular automata. This library is based on JAX and can thus utilize modern computation hardware better, leading to significant performance increases in some cases as the authors show on a few common examples for CA applications.

**Strengths:**

The issue the library tackles is impactful. CAs play a huge role in various areas of research and having efficient implementations is, of course, key to experimenters. Dramatic improvements in performance have been known to enable a way broader research community in other fields.

The library also unifies various scenarios in which CAs are used that have been fragmented over multiple platforms.

The foundation in JAX also seems to allow for comparatively easy integration with other approaches in the field of AI. (A discussion of that point might be helpful.)

The libraries capabilities are demonstrated on a well-selected range of applications.

Various interesting novel applications are defined and well presented in the paper.

**Weaknesses:**

The paper stresses some standard library features (like documentation or examples) too much compared to more research-relevant parts.

The performance comparison should be broader. Various libraries exist. Why were only two tested in only very specific cases?

The performance difference to TensorFlow is left unexplained. TF should use hardware acceleration as well, so what is happening there?

The figures are not properly referenced in the text.

Some formal problems exist:
- superfluous ")" in "Section 2)" on page 4
- ill-formatted citation of "Mordvintsev et al." on page 6
- many wrong quotation marks on pages 8-10
- "doesn't" on page 8
- "a NCA" on page 8
- superfluous comma "training set," page 10
- ill-formatted reference to Appendix A on page 10

**Questions:**

None.

---

> ### Author Response · Authors · 2024-11-21
>
> We appreciate your comments and the opportunity to address your concerns. We address each point below.
>
> ---
>
> > The paper stresses some standard library features (like documentation or examples) too much compared to more research-relevant parts.
>
> Thank you for this feedback. As stated in both our abstract and introduction, the goal of this paper is to introduce this new library and highlight its benefits to the research community. This is why the paper emphasizes certain aspects of the library like documentation and examples, as we believe they are essential for successful adoption by researchers.
>
> The more research-relevant parts of the paper, particularly the experiments in Section 6, were specifically designed to demonstrate and showcase the library's capabilities: "_We demonstrate CAX's potential to accelerate research by presenting a collection of three novel cellular automata experiments, each implemented in just a few lines of code thanks to the library's modular architecture_". As such, these experiments are not intended as full-fledged research contributions. Given that this paper's primary contribution is the library itself rather than the experiments, we chose to prioritize explaining the framework's architecture and design principles.
>
> That being said, we agree with the reviewer that the research-relevant parts are also very valuable and deserve thorough treatment. Therefore, we have expanded Section 3 to provide a more comprehensive explanation of the framework's foundations and innovative aspects.
>
>
> > The performance comparison should be broader. Various libraries exist. Why were only two tested in only very specific cases?
>
> Thank you for this comment. While you are right that a few CA libraries exist, to the best of our knowledge, **CAX is actually the first hardware-accelerated cellular automata library**.
>
> Existing hardware-accelerated NCA implementations are always tied to specific research papers and are not designed as general-purpose libraries. Therefore, in the paper, the NCA training benchmark compares CAX against these individual paper implementations, rather than against another general-purpose NCA library, since no such library exists. These paper implementations are typically focused on reproducing specific experiments and cannot be easily adapted for other NCA tasks or research directions. The fragmented nature of existing NCA implementations makes comprehensive library comparisons impossible.
> While non-hardware-accelerated libraries like CellPyLib and Golly exist, as outlined in our related work section, they serve different purposes. For instance, Golly is a cross-platform cellular automata simulator but isn't designed for hardware acceleration or integration with modern machine learning frameworks.
>
> In the paper, we compare CAX with CellPyLib on the two most canonical cellular automata - Game of Life and Elementary CA to give users a concrete indication of the speed-ups they can expect.
>
> However, we believe that a broader comparison with non-hardware-accelerated libraries would not be informative as they serve a different purpose and would not be able to compete in terms of runtime performance due to the lack of hardware acceleration. Moreover, common CA libraries are usually limited to two-dimensional and discrete cellular automata and not designed for training neural cellular automata.
>
>
> > The performance difference to TensorFlow is left unexplained. TF should use hardware acceleration as well, so what is happening there?
>
> The performance difference between CAX and TensorFlow implementations stems largely from our optimized handling of the Sample Pool mechanism, which was originally introduced in the Growing Neural Cellular Automata paper by Mordvintsev et al.
>
> In CAX, we provide an implementation of this mechanism that has been developed with a strong attention to runtime optimization. It avoids costly data transfers between CPU and GPU, and enables efficient batched operations through JAX's `vmap` and compilation through `jit`.
>
> Thank you for pointing out that this was not explained in the text. We have changed the manuscript to make this point clearer.
>
>
> > The figures are not properly referenced in the text. Some formal problems exist...
>
> We sincerely appreciate your meticulous attention to typographical and formal issues. Thank you for helping us improve the paper!
>
> ---
>
> We hope we have answered all your questions and concerns. Please do not hesitate to bring up any additional questions.

---

### Author Response · Authors · 2024-11-21

Dear Reviewers,

We sincerely thank you all for your thorough and constructive feedback. We appreciate the time spent reviewing our work.

We are grateful for the positive reception of several key aspects of our work. Multiple reviewers highlighted CAX's potential impact on cellular automata research, with _5JEh_ noting that "dramatic improvements in performance have been known to enable a way broader research community" and _XZsB_ emphasizing how CAX will "accelerate the study of emergence and self-organization in cellular automata." _BH4L_ and _7WC1_ appreciated the presentation and writing of the paper, as well as the timeliness of GPU acceleration through JAX. Finally, _5JEh_ valued how CAX "unifies various scenarios in which CAs are used that have been fragmented over multiple platforms."

To address the main concerns raised in the reviews, we have made several improvements.
- We have added the ViTCA experiment to our anonymous repository to demonstrate CAX's ability to implement attention-based neural cellular automata, showcasing further the framework's flexibility in incorporating advanced architectures.
- We have also clarified CAX's position within the machine learning context in the introduction and strengthened the emphasis on the research-relevant aspects of the paper, particularly the controllable CA framework and the perceive/update architecture that constitute the core of CAX. These elements enable seamless integration with modern deep learning techniques while maintaining the cellular automata paradigm.

Thank you again for your valuable feedback.

---

### Meta-Review · Area_Chair_K654 · 2024-12-17

**Metareview:**

**Summary:**
This paper proposes CAX, which is an open-source python library for cell automaton (CA) research. It makes use of GPU acceleration provided by the JAX library, which would enable significant speeding-up of CA simulations.

**Strengths:**
The library has implemented several different types of CA, ranging from classic ones (such as elementary CA and Conway's game of life), through a variety of neural CAs, to quite new instances (those showcased in Section 5). Massive speedup against two benchmarks has also been demonstrated in Section 3.2.1.

**Weaknesses:**
Some reviewers (5JEh, BH4L, 7WC1) raised concern about relative weakness of the research-relevant contents of this paper compared with its documentational nature.

**Reasons:**
Despite the concern on research-relevant contents of this paper, all the reviewers agreed that the strengths of this paper outweigh the weakness, recommending its acceptance.

**Additional Comments On Reviewer Discussion:**

All the reviewers rated this paper high above the acceptance threshold. The authors addressed those concerns raised by the reviewers appropriately in their rebuttal. What is notable is that, in response to Reviewer BH4L's suggestion, they managed to implement the vision transformer cellular automata (ViTCA) (Mattie, Nowrouzezahrai, Pal, "Attention-based neural cellular automata," NeurIPS, 8174-8186, 2022) during the brief period of the author rebuttal, demonstrating the utility of the library.

---

### Decision · Program_Chairs · 2025-01-22

Accept (Oral)